# Accessibility of Online Resources for Associations Providing Services to People with Brain Injuries in Covid-19 Pandemic

**DOI:** 10.3390/ijerph182312609

**Published:** 2021-11-30

**Authors:** Nolwenn Lapierre, Olivier Piquer, Erik Celikovic, François Routhier, Julie Ruel, Marie-Eve Lamontagne

**Affiliations:** 1Centre Interdisciplinaire de Recherche en Réadaptation et en Intégration Sociale, Université Laval, Québec, QC G1V 0A6, Canada; nolwenn.lapierre.1@ulaval.ca (N.L.); olivier.piquer.1@ulaval.ca (O.P.); erik.celikovic.1@ulaval.ca (E.C.); francois.routhier@rea.ulaval.ca (F.R.); 2Chaire Interdisciplinaire de Recherche en Littératie (CIRLI), Institut Universitaire en Déficience Intellectuelle et en Trouble du Spectre de L’autisme, Trois-Rivières, QC G8Z 3T1, Canada; julie.ruel@uqo.ca

**Keywords:** accessibility, disability, digital resources, Covid-19

## Abstract

Background. Since the Covid-19 pandemic, many community-based services for people with traumatic brain injury (TBI) have been moved online, which may have hindered their accessibility. The study aims to assess the accessibility of online information and resources dedicated to people with TBI. Methods. The websites of 14 organizations offering information and resources to people with TBI in Quebec were evaluated. Two co-authors independently evaluated one page of each website and compared their results. Descriptive statistical analyses were performed. Results. The average accessibility score of the 14 websites evaluated was 54% with a standard deviation of 16%. Website design and writing were the most accessible aspects (72.3%). Only two out of the 14 websites (14%) presented multimedia content. This category presented the most barriers to accessibility with a score of 42%. Regarding images, they reached an accessibility score of 46%. Their main shortcoming was the absence of a caption. Conclusion. This study highlights accessibility issues specific to people with TBI to access online resources and identifies specific areas of improvement. The results of this study provide community organizations with avenues of improvement to make their online resources more accessible to people with TBI and may therefore lead to improved community practices.

## 1. Introduction

In Quebec, there are 17,400 new cases of traumatic brain injury (TBI) each year [1]. Among the difficulties that this population encounters, one will note physical, cognitive (e.g., deficit of attention), or psychological challenges [1,2,3]. These disabilities affect quality of life [1], independence and social interactions [2,4]. Thus, many individuals with TBI require long-term support to be able to participate in their community after the end of their rehabilitation [5]. This support is often offered by community-based organizations [6,7]; they provide various services, mainly on-site such as occupational activities, home support, social support or respite programs for caregivers [3].

However, since March 2020, the beginning of the Covid-19 pandemic in Quebec, many of the services offered by the associations have had to be urgently transferred online or remotely, endangering the indispensable accessibility of community-based services for individuals with a brain injury. This change highlighted a variability among those with TBI regarding the use of digital resources, such as websites [4]. In fact, while some individuals accessed online services with ease, others faced challenges in doing so and thus were unable to get the services they needed. In rehabilitation contexts, telerehabilitation is generally feasible (1) and satisfying (2); however, there is less evidence of optimal fit between individuals with TBI and internet in natural, undirected settings. The proportion of TBI individuals using the Internet is high (74%) but remains lower than the general population (84%) [5]. While the difficulties in engaging this population in online resource use might be indicative of a low basic and digital literacy [6], another important hypothesis to this difficult situation is accessibility issues of web resources [7].

Accessibility is, along with the availability, acceptability, affordability, usability, and adaptability, one of the dimensions that allow for access of a given resource [7]. The Human Development Model—Disability Creation Process (HDM-DCP) stipulates that environment, including virtual environment, interact with individuals’ abilities and life habits to create handicap situation or to allow for social participation [7]. Accessibility to the web is thus one of the many potential strategies to avoid handicap situation. Web accessibility is defined as “the possibility that a product, environment or service available through the Internet can be used in equality, safety and comfort by all people, and especially for those with disabilities” [7] (p. 92). While the accessibility of online resources for a given population is crucial to their use, the context of the Covid-19 pandemic dramatically highlighted the importance of this dimension for people with TBI. The accessibility challenges faced when using online services in pandemic times can particularly be felt by individuals with TBI living in remote areas or experimenting challenges to reach in-person services because of fatigue or lack of transportation. Despite its importance for access to service, little is known about accessibility of online resources for people with TBI.

Accessibility guidelines (web content accessibility guidelines, WCAG) have been established [8] to allow for universal accessibility of online resources. These guidelines include recommendations both for the content and the format of online resources (e.g., guidelines for text alternatives, support to user’s navigation) and define success criteria for accessibility to a wide range of people with disabilities [9]. Detailed indicators have also been developed by Ruel et al. [10] to evaluate the accessibility of communication tools, including online resources. Similar to the WCAG, Ruel et al.’s indicators are not specific to individuals with TBI; however, they do provide eight categories to assess, with detailed indicators for all of them [10]. These categories are the following: (1) navigation (including, for example, the navigation menu or the adaptability of the website), (2) conception and design (e.g., structure of the website page), (3) images, (4) multimedia content, (5) digital files, (6) forms and questionnaires, (7) user protection and captcha and (8) slideshow.

Automatic tools also exist to evaluate website accessibility and identify access barriers. Most of existing automatic evaluation tools are based on the web content accessibility guidelines (WCAG) [11]. However, the results of automatic tools for accessibility evaluations are often described as lacking essential information, whereas manual evaluation are subjective and lack feasibility because of the extent of the WCAG [8]. Another limitation of automatic tools is that they do not always identify accessibility barriers specific to certain types of disability [8]. Therefore, Orozco et al. (2016) suggest integrating both automatic and manual evaluation and defined a framework for the evaluation of the web accessibility for people with disability [8]. Despite the availability of accessibility evaluation tools, up to date there is no existing evaluation of accessibility of online resources for individuals living with a TBI. This gap needs to be filled to better organize the online service offer for this population in and after the Covid-19 pandemic.

The aim of the study is to assess the accessibility of online information and resources offered to members of associations for people living with brain injury in Québec (Canada).

## 2. Materials and Methods

### 2.1. Research Design

The study follows a transversal quantitative design with a descriptive approach [12].

### 2.2. Participants and Recruitment

The target population includes the 14 websites of the provincial board (*n* = 1) and associations (*n* = 13) delivering online resources to people with TBI from the province of Quebec. The inclusion criteria were: (1) to be an association providing resources to people TBI in Québec (Canada), (2) having a public website providing online resources to individual with brain injury. Associations were identified with the support of the Regroupement des associations de personnes traumatisées craniocérébrales du Québec (ConnectionTCC.QC, https://www.connexiontccqc.ca/ (accessed on 22 November 2021)), regrouping 13 associations through the province of Quebec.

### 2.3. Data Collection

The data collection was based on Orozco et al. (2016) framework [8]. This approach based on the WCAG [11] consists of five stages: The assessment of the online resources followed the five steps described by Orozco et al. (2016): (1) analysis and the characterization of the population, (2) definition of indicators for evaluation, (3) definition of heuristics, (4) implementation of accessibility evaluation, (5) analysis of results. The extraction of the relevant characteristics of the population for the indicators was done by two co-authors (OP and MEL) who have clinical and research experience with individuals with TBI. This analysis led to the definition of indicators and measures for evaluation based on Ruel et al. (2018) “Listes de vérification pour la conception de sites web et supports numériques” (Checklists for website and digital media design) [10,13]. The checklist has been operationalized into quantitative indicators. Objective appraisal of each concept has gathered in an assessment grid (Table 1). The assessment grid was organized into the seven of Ruel et al.’s categories: (1) navigation, (2) design and writing, (3) image, (4) multimedia content, (5) digital files, (6) forms and questionnaire, (7) protection and CAPTCHA and an additional category: (8) Facebook content.

Following the recommendation of Orozco et al. (2016), the websites were analyzed both automatically and manually [8]. For the automatic evaluation, the websites and Facebook pages of the included associations were assessed using the automatic assessment the Web Accessibility Evaluation Tool (WAVE, https://wave.webaim.org (accessed on 22 November 2021)) [14]. The WAVE is based on the principles of the WCAG 2.1 [15]. To evaluate the literacy needed to understand the content of the websites and Facebook pages, their content has been analyzed with Scolarius (https://www.scolarius.com/ (accessed on 22 November 2021)). This automatic tool defines which education level is required to read and understand a web content: (1) primary, (2) secondary, (3) college, (4) university, (5) mastery of the subject. For the item 22 (the sans serif font is used, Table 1) of the category design and writing, a sans serif font was defined following the guidelines of the Government of Quebec [16]. The browser extension Font Finder [17] was used to assess the font information required to address items 23 (The font used is non-condensed) and 24 (The font used measures at least 16 pixels) of the grid.

### 2.4. Procedure

The data collection method was applied to the websites and Facebook pages of the included associations between March 2021 and August 2021. The authors applied the evaluation grids, that encompass both automatic (WCAG) and manual assessment. For each resource, two authors (OP and NL) assessed independently one page of the websites and compared their output to reach an agreement. Disagreements between the two co-authors were either resolved by consensus or by another co-author (MEL). After this training evaluation where a perfect inter-rater reliability was reached (e.g., the two evaluators provided same scores), one co-author assessed a maximum of four other pages per website (to reach a total of 5 pages per website). The same pages (e.g., welcome page, programming, contact form) were chosen for the evaluation of each website.

### 2.5. Analysis

Descriptive analyses were performed; for non-parametric data, the median and interquartile ranges (IQR) were calculated based on the results of the automatic and manual evaluation. To determine the accessibility score for each website, the score for each item (see Table 1) was used to calculate the success percentage (0% not at all accessible; 100% fully accessible) for each category of the grid. The medians of accessibility scores were calculated per category for all websites.

## 3. Results

### 3.1. Included Online Information and Resources

The websites of provincial board and the 13 associations dedicated to people with TBI and representing approximately 2000 members across the province of Quebec (Canada) were available and all included in this study (100% representation). Their members were individuals with TBI from various levels of severity (from mild to severe). The 14 websites and the content of their Facebook pages were assessed for accessibility. The median score of accessibility for these resources was 55% (IQR = 24.4). Details about the accessibility score per assessment category are presented in Figure 1.

### 3.2. Navigation

Regarding the navigation of their website, the included websites reached a median score of 46% (IQR = 9.6). The items that were the most lacking across websites were a site map (skeleton of the site) accessible on the home page and a warning message when a new tab or window is automatically opened; none of the fourteen websites presented these features. However, some elements of navigation accessibility were successful for all 14 associations: (1) reaching a page from the menu required no more than 3 clicks, (2) the navigation area was separated from the content on all pages of the site and (3) the menu bars were horizontal (not vertical). Regarding the level of language need to understand the content of the included websites, *n* = 9/14 (65%) websites were using a language level above junior high.

### 3.3. Design and Writing

Among all the accessibility categories, the design and writing of the website was the most accessible one with a median score of 72% (IQR= 5.2). Two accessibility features of the website design and the writing were lacking in each of the 14 websites: a “Frequently asked questions” section and an email address for dedicated technical support. On the contrary, 13/35 (37%) of the design and writing features were present in all 14 websites (e.g., the text was divided into paragraphs, the text was not justified, the font used was uncondensed).

### 3.4. Image

Images presented in the websites reached an accessibility score of 50% (IQR = 33.3). Two out of the fourteen associations (14%) did not present any images in the analyzed pages of their websites. When present, images (*n* = 12 websites, 100%) lacked captions. Likewise, half of the website images presented lacked diversity in the people represented (i.e., in terms of age, ethnicity). The feature favoring accessibility the most in the websites was a textual support to the proposed image (*n* = 9/12 websites, 75%).

### 3.5. Multimedia Content

Only two associations (14%) offered multimedia contents to their users in their websites. These contents reached a median score of 42% (IQR = 11.9), making this category the most problematic in terms of accessibility. Two aspects received the worst scores: (1) the proposed videos exceeded 2 min and (2) no textual alternative to the video was proposed. On the contrary, both websites including multimedia contents in their websites offered links to go to the external sources of the video or visible volume control knob to facilitate the accessibility of their content.

### 3.6. Digital Files

Five associations (36%) provided digital files to their members through their websites. The median accessibility score for this category was 33% (IQR = 16.7). Four out of five websites had digital files with a summary including clickable links. In addition, the different hierarchical levels were obvious for these five websites.

### 3.7. Forms and Questionnaire

Half of the included associations offered forms and questionnaires on their websites, mainly to provide members with an easy way to contact them. The median accessibility score of these forms and questionnaire was 55% (IQR = 9.4). The most accessible aspects of these forms were that a title was identified, the forms respected the visual interface of the site, they could be opened in all major browsers and that their format was identical for the whole form in *n* = 9 (100%) websites. The aspects that most hindered the accessibility of website forms were that the false entries were not reported as they occurred in *n* = 9 (100%) and that no e-mail confirming form submission has been sent to the user in *n* = 8/9 (89%)

### 3.8. Protection and CAPTCHA

The median score for the accessibility for the data protection and the CAPTCHA on the website pages was 50% (IQR = 18.8). In 13 out of the 14 websites, no validation was required by the CAPTCHA, which was the most accessible features across website in this category. However, most of the websites (*n* = 12/14; 86%) did not provide a privacy policy.

### 3.9. Facebook Content

Twelve out of the fourteen associations (86%) had a Facebook page. The median score for the level of language needed to understand the content was 65% (IQR= 21.3), which means that for most of the Facebook pages, the published content mainly required a secondary grade level or lower.

## 4. Discussion

In this transversal quantitative study, the online content of fourteen associations delivering resources to people with TBI has been evaluated for their accessibility. We overall found moderate level of accessibility, which may highlight that some of the TBI association members, independently of their digital and basic literacy level, might face challenges in reaching services they needed.

The study of Brunner et al. (2017) highlights that information and communication technologies are frequently used for individuals with TBI’s rehabilitation [18]. However, the current study shows that the resources available online for people with TBI are moderately accessible (accessibility score of 55%) which is consistent with the accessibility concerns reported the participants of Morrow et al. (2021) study [4]. Morrow et al. (2021) reported that accessibility, safety and usability remain important barriers for their use of online resources [4], reinforcing the importance of developing solutions to improve the accessibility of online resources.

Individuals with TBI have different use patterns for social media, videoconferencing platforms and general internet than non-injured individuals [4]. They need simplified content, visually accessible with low complexity. Thus, accessibility evaluation of online resources for individuals with TBI is very important and should include criteria specific to this population.

### 4.1. Developing an Assessment Grid Specific to Individuals with TBI

The universal accessibility standards built into the WCAG do not cover all the needs of people with TBI. Special attention must therefore be paid by associations providing services to people with TBI. Hence the need to develop an analysis grid such as the one presented in this study. Applying Orozco et al. (2016) framework enabled us to create an assessment grid considering international guidelines on web accessibility and recent work on website and digital media accessibility [10,11]. This assessment grid addresses a gap in the literature that has been highlighted in previous studies about online resources for individuals with TBI. However, some authors recommend conducting a three-step evaluation of online resource accessibility [19]: (1) by users themselves; (2) by an evaluator or expert in the field of web accessibility, who assesses the site’s compliance based on recognized criteria; (3) by automated tools. Following a user-centered approach [20], to improve the accessibility of these resources, exploring the experience of individual with TBI when accessing resources through qualitative research might be of interest to improve resource accessibility and will be the subject of a future publication. Adapted methods, such as the Think-Aloud [21] or online Photovoice [22], could provide unique insights into the use of online resources for people with disability, such as people with TBI.

### 4.2. Barriers to Accessibility

The navigation of the included online resources presented many accessibility barriers. Facilitating this specific accessibility aspect may address an issue raised in the study of Brunner et al. (2019): participants mentioned having to proceed by trial and errors in their use of social media [23]. Navigation is particularly of importance knowing that individuals with TBI report lacking support from their relatives and health professionals in their use of these online resources [23].

Among the barriers to accessibility identified in the literature, the reading age required to understand the content of the online resources is often raised; the study of Manivannan et al. (2021), for instance, reports that less than 30% of the websites included in their study followed the recommended reading age for people with TBI [24]. The present study confirms this concern with most of the included online resources using a language level above junior high school according to Scolarius [25]. To be intelligible by all people with disability, including individuals with TBI, the content of the websites and Facebook pages should be of primary or secondary level [26,27]. This means that a special effort must be made by associations to provide content in language that is accessible to people with TBI.

### 4.3. Study Limitations

The study’s main limitation is that only the online content of Quebec associations was evaluated, which may limit the generalization of the results. The terminology used in the literature for the accessibility evaluation of online resources dedicated to individuals with TBI is not yet consensual, which leads to difficulties in reaching consensus on the evaluation of certain grid items. However, applying Orozco et al.’s framework, involving experts in TBI and technology accessibility in the research team, as well as validating the evaluation of each resource by at least two team members may have mitigated this bias.

## 5. Conclusions

The accessibility of existing online resources for people with TBI is moderate. This study proposes a grid to specifically assess the accessibility of online resources for individual with TBI. When applied to the online resources of associations from Quebec, this assessment highlights accessibility issues specific to people with TBI to access online resources and identifies specific areas of improvement. If these improvements are implemented, individuals with TBI may have better access to community-based resources and services supporting their engagement in their communities. Although further research is needed in both assessment tool and online resources accessibility, the results of this study provide community organizations with avenues of improvement to make their online resources more accessible to people with TBI, which may therefore lead to improved community practices.

## Figures and Tables

**Figure 1 ijerph-18-12609-f001:**
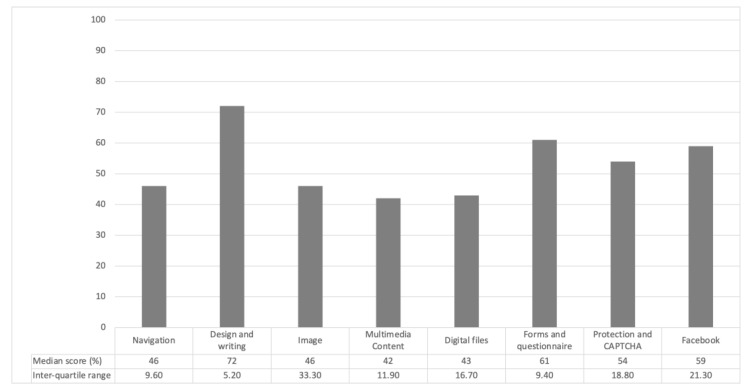
Median score and inter-quartile range per assessment category for the 14 associations.

**Table 1 ijerph-18-12609-t001:** Assessment grid.

Evaluation Grids Sections and Items
1. Navigation
1.1. Site arborescence
The site map (skeleton of the site) is accessible on the home page
Presence of thread on each page of the site
Reaching a page from the menu requires a maximum of 3 clicks
The sub-menus offer a maximum of 5 choices
Navigation is consistent
The menu is identical on each page of the site
One or more navigation options are present on all pages of the site
One or more basic options (e.g., previous, following, home) used for navigation are present on each page of the site
One or more basic navigation options are easily visible on each page of the site
One or more basic navigational options are "just a click away" on each page of the site
One or more basic navigation options are always in the same place on each page of the site
The TAB key is used to move around the site
Available keyboard shortcuts are presented
A page on the adaptations of the site is available in the menu
1.2. Navigation menu
The navigation area is separated from the content on all pages of the site
The menu bar is horizontal or circular
The most important menu items are located on the left
The menu buttons are recognizable
One or more scrolling help options are available (e.g., "return to top of page")
The footer provides links to one or more level 1 menu items
A print tool is available at the top or side of the page
A "send by email" tool is available at the top or side of the page
A character size tool is available at the top or on the side of all pages of the site
One or more navigation tools are presented in a single menu
The banner (top band of the site) has a maximum of 8 interactive tools (buttons, links, icons)
1.3. Search function
A search engine is proposed on each page of the site
The search engine is tolerant (adapts to mistakes)
The search engine is intelligent (automatically corrects mistakes, suggests related terms, key words…)
There is more than one search option available (by date, by subject, by author…)
1.4. Hyperlinks
Hyperlinks are underlined
The names of the linked documents are explicit (official title related to the content)
A message warns the user of the opening of a new tab or a new window (tooltip or logo)
The document opens in a new tab or browser window
2. Design and writing
2.1. Structure of the web page
The content hierarchy is clearly identified in the page code (identified in the WAVE)
The coherence of the hierarchy is respected throughout the site
The text is divided into paragraphs
The essential information is located above the waterline on each page
The content is located in the center of the page (there are margins all around)
Text lines are 80 characters or less
Line spacing is at least 1.5 times the text size
The space between paragraphs is at least 1.5 times the size of the line spacing
Text is left aligned
The text is not justified
There is no indentation at the beginning of the paragraph
Bulleted or numbered lists are used to facilitate reading
The presentation style of the site is consistent
The presentation style of the site is homogeneous
2.2. Structure and content of the information
Pages contain a maximum of 110 words
The sentences contain a maximum of 20 words.
Paragraphs consist of a maximum of 6 lines
The titles are composed of a maximum of 4 words
The language level used corresponds to the first cycle of secondary education according to Sco- larius.
There is a "frequently asked questions" section
2.3. Visual aspects
The font used is sans serif
The font used is uncondensed
The font used measures at least 16 pixels
Zoom is possible up to 200%.
When the text reaches 200%, it is not necessary to scroll horizontally to read it
The contrast of the site reaches at least the ration of 3:1 (WAVE)
The choice of background color is at least A (WAVE)
2.4. Update
The date of the last update is displayed
The date of the last update is located near the title of the page
2.5. Site Transmitter
The name of the site appears on all pages
The name of the site is always located in the same place on the pages
The organization’s logo appears on all pages
The organization’s team is presented on the site
A contact person is available
The email of a person dedicated to the technical side of the site is identified
3. Images
The text supports the proposed image
The audiences represented in the images are diverse (gender, colors, limitations…)
The images are accompanied by an explanatory text or a legend
4. Multimedia content
The video can be played by different players
Links are available to external sources of the video (e.g., Youtube)
The video can be downloaded
The duration of the videos is a maximum of 2 minutes
Video control buttons are available when the player opens
The volume control knob is visible
Visual aids accompany the video
A textual alternative to the video is proposed
Subtitling of the video is available
The subtitle speed setting is available
The size of the subtitles can be adapted
The contrast of the subtitles can be adapted
Subtitles are understandable
5. Digital files
The different hierarchical levels are obvious
A summary with clickable links is available
Tables, diagrams, illustrations or images are described (caption or explanatory text)
6. Forms and questionnaires
6.1. Attention to the user
The title is identified
An explanatory introduction is available
The instructions (titles) accompany the items
Tooltips give explanations to help the user to fill the form
A link is available to an explanatory appendix of the form
A single click allows you to go back and forth between the appendix and the form
False entries are reported as they occur
Errors are explained in text form
Errors can be corrected
Suggestions for corrections are offered
It is possible to go back to previous answers
There is no time limit to complete the form
It is possible to save the form to complete it later
An email confirms to the user that they have completed and submitted their form
An email reminds the user that the form is incomplete if applicable
There is a downloadable version of the form
6.2. Presentation
The form respects the visual interface of the site
The form offers closed questions
The form offers check boxes (to be validated in a set of choices)
The form offers drop-down lists
An answer of type "other" or "don’t know" is available in the answer choices
The zones (transmitters / users) are differentiated
The number of characters for the answer is unlimited
For an updated form, the changes are clearly identified
An alternative version (audio for example) is available
Abbreviations are explained
The form can be opened in all major browsers (e.g., Firefox, Safari)
6.3. Content and structure
The language used corresponds to the level of the target population (Scolarius)
The form offers examples of answers
The questions are grouped by theme
The fields of the form are described
The most important questions are at the beginning of the form
Questions are identified (numbered, lettered…)
The different sections of the form are identified by titles
The format is identical for the whole form
A maximum of 5 choices is offered in the multiple-choice questions
The name of the documents to be attached is indicated
The procedure for the transmission of attachments is indicated
A confirmation of sending or receiving is sent to the user
7. Protection and Captcha
7.1. Protection of Internet users
Privacy settings are enabled (secure pages)
No confidential information is requested
A privacy policy is available
7.2. Captchas
No validation is required by the CAPTCHA
8. Facebook content.

## Data Availability

The data presented in this study are available on request from the corresponding author. The data are not publicly available du to ethical reason.

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
