# Peer review of "Accessibility of Online Resources for Associations Providing Services to People with Brain Injuries in Covid-19 Pandemic"

_ijerph, 2021, doi:10.3390/ijerph182312609_

Round 1

Reviewer 1 Report

The strength of the paper is its focus on a topic of interest. The Covid-19 pandemic has brought about the largest increase in digitalization of the rehabilitation services seen so far and it is important to have available internet resources for patients with TBI.  It is overt that the online resources should be closely evaluated regarding content and user acceptability. Because no two people with TBI have the same needs (there is great heterogeneity in severity and outcome), website designers and developers should tailor their accessibility efforts toward the widest possible audience with TBI. TBI patients often take longer to think about their decisions and they are also more easily distracted by designs or navigational flows that are complicated or cluttered.

I have a few minor comments:

  1. When reporting the percentages I would consider to "round more" and report the percentages as whole numbers.
  2. Line 246; abbreviation CBD-what does it stand for? Chronic brain disorder?
  3. Line 255-258: " Following a user-centered approach , to improve the accessibility of these reources, exploring the exprience of individual with TBI when accessing resources through qualitative research might be of interest to improve resource accessibility and will be the subject of a future publication." Indeed, the ultimate quality criteria for the services including online resources are to what extent the users’ needs are met. Thus, the voices of the disability population are of major importance in the evaluation of online resources and content of rehabilitation services during and after the Covid-19 pandemic. I look forward to the future publication on this topic.

Author Response

We thank the reviewer for his/ her comments.

 1 -We rounded the percentages as suggested.

2 – We apologize for the confusion. The right term that should be used is TBI, traumatic brain injury. We corrected the text accordingly.

3-We thanks the reviewers for his/her interest. These works are currently under analysis and would hope

Reviewer 2 Report

The topic is interesting. The structure of the paper is clear, but I have minor iusses about theoretical rationale underlying of your study and presentation of results. I feel that these issues should be addressed in a minor revision. My concerns are listed below.

  1. The Introduction should be improved. A theoretical framework should be provided to justify the rationale of your study.
  2. What are community-based services for individuals with a brain injury that you refer in your paper? 

    3.  Given that the topic of your study is the accessibility of online resources, a discussion about tele-rehabiliation services for people with disability and satisfactions of user is necessary. (Please, see the studies of Dovigo et al. 2021 and Caprì et al. 2020).

    4. What was the inter-reliability for the two independent observers?

    5. Please, report median and IQR for each parameter in a table

    6. Figure 1 must be improved. Report the percentage with a range of 10.

Author Response

We thanks the reviewers for his / her suggestions.

1-We add on page 2 that "The Human Development Model – Disability Creation Process (HDM-DCP) stipulates that Environment, including virtual environment, interact with individuals’ abilities and life habits to create handicap situation or to allow for social participation[7]. Accessibility to the web is thus one of the many potential strategies to avoid handicap situation. "

2-We already mentionned on page 2 that This support is often offered by community-based organizations [6, 7]; they provide various services, mainly on-site such as occupational activities, home support, social support or respite programs for caregivers [3].  We wondered if further detailled descriptions of these services are required as it is not the main objective of this paper. We can add details if the reviewers and editor feel it required. 

3-We agree that some arguments can be add about telerehabilitation of adults with TBI can be add, but it have limits since these interventions are usually much more structured and specialized than what can be offered by associations through their low tech plateform, and the individuals are not in the chronic phase of their condition. We however add that : In rehabilitation contexts, telerehabilitation is generally feasible(1) and satisfaying(2); however, there less evidence of optimal fit between invidiuals with TBI and internet in natural, undirected settings. 

4- We added the IQR to Figure 1, that already rapported median score in a visual way; we also modify Figure 1 according to the reviewer suggestions. 

Reviewer 3 Report

Dear authors, congratulations for your manuscript and for the novelty and originality of it. It has an important significance for the improve of accessibility of online resources to people with brain injuries. 

I don't have any suggestions for the improvement of you manuscript, but I give you the challenge to study the perspective of the users, analysing the patients experience in using this resources, that seem to have, in general good acessibility scores in the different dimensions! 

Looking forward to read more from your team. 

Author Response

We thanks the reviewer for his / her encouragements; our works about users perspective are currently under analysis.